# Emergent helical edge states in a hybridized three-dimensional topological insulator

Su Kong Chong [1] ✉, Lizhe Liu [2], Kenji Watanabe [3], Takashi Taniguchi [3], Taylor D. Sparks [2], Feng Liu [2] & Vikram V. Deshpande [1] ✉

As the thickness of a three-dimensional (3D) topological insulator (TI) becomes comparable to the penetration depth of surface states, quantum tunneling between surfaces turns their gapless Dirac electronic structure into a gapped spectrum. Whether the surface hybridization gap can host topological edge states is still an open question. Herein, we provide transport evidence of 2D topological states in the quantum tunneling regime of a bulk insulating 3D TI BiSbTeSe$_2$. Different from its trivial insulating phase, this 2D topological state exhibits a finite longitudinal conductance at ~2e$^2$/h when the Fermi level is aligned within the surface gap, indicating an emergent quantum spin Hall (QSH) state. The transition from the QSH to quantum Hall (QH) state in a transverse magnetic field further supports the existence of this distinguished 2D topological phase. In addition, we demonstrate a second route to realize the 2D topological state via surface gap-closing and topological phase transition mechanism mediated by a transverse electric field. The experimental realization of the 2D topological phase in a 3D TI enriches its phase diagram and marks an important step toward functionalized topological quantum devices.

Despite the well-studied topological surface states in 3D TIs[1], their peculiar gapped surface states in the presence of broken symmetry have still been a subject of intense studies for exotic topological quantum states. Even in the absence of symmetry breaking, the strong coupling between the topological surface states in a 3D TI can be a medium to realize interesting states, e.g. the quantum spin Hall (QSH) effect[2–4]. Such 2D TIs host a surface gap with inverted bands initiated via hybridization of the surface states of the parent 3D TIs. From the theoretical perspective, the gapped surface spectrum exhibits an oscillatory behavior alternating between topologically trivial and nontrivial 2D states as a function of layer thickness[2–4]. Although the hybridization gap has been systematically probed in prototypical 3D TIs Bi$_2$Se$_3$ by angle-resolved photoemission spectroscopy (ARPES)[5,6], and Sb$_2$Te$_3$ by scanning tunneling microscopy[7], the anticipated 2D TI phases and their QSH states await to be confirmed electrically in device measurements. Previous transport measurements in the hybridization regime were restricted by disorder and inhomogeneity in TI thin films

preventing the observation of the surface hybridization gap[8,9]. Also, the crystal quality of the TI films limited the mobility which constrained the mean free path to microscopic length scales[10]. From the technical point of view, electrical measurements can resolve the prevailing small energy gap scales beyond the resolution limit of ARPES[5,6].

Band inversion through a gap-closing and reopening mechanism is a hallmark of a topological phase transition (TPT)[1]. In addition to the layer-dependent topological phases varying with the thickness of 3D TIs, a TPT can be controllably induced through band distortions by in-plane or out-of-plane magnetic field[11,12], strain, or pressure[13,14], and electric field[7,15–18]. The external in-plane magnetic field can induce gap-closing by oppositely shifting the two surface bands[11]. However, a strong magnetic field is required to fully close the surface gap which is impractical for most applications. Another mechanism with which to transform the band topology is by distorting the crystal lattices, such as strain or pressure-induced TPTs in ZrTe$_5$[13] and BiTeI[14], respectively. Finally, the most preferred route is a TPT driven by perpendicular

[1]Department of Physics and Astronomy, University of Utah, Salt Lake City, UT 84112, USA. [2]Department of Materials Science and Engineering, University of Utah, Salt Lake City, UT 84112, USA. [3]National Institute for Material Science, Tsukuba, Japan. ✉e-mail: sukong.chong@utah.edu; vdesh@physics.utah.edu

electric fields, which can realize a functional all-electrical topological switch between normal and inverted gap states by controlling the electric potential between top and bottom surfaces[16,17].

In this work, we study the magneto-electrical transport properties of the ultrathin BiSbTeSe₂ 3D TIs in the intersurface hybridization regime. We identify the normal insulator and 2D topological insulator phases as indicated by their distinct responses to temperature and magnetic field. We further demonstrate that these trivial and topological insulating phases can be reversibly tuned by a perpendicular electric field through a gap-closing mechanism. The highly tunable topological phases enrich the phase diagram of 3D TIs in the 2D thin limit.

## Results

### Topological phase diagram

We first perform the first-principles calculations for the intersurface hybridized $Bi_{0.7}Sb_{1.3}Te_{1.05}Se_{1.95}$ (hBSTS) 3D TI. The hybridization between the top and bottom surface states of the BSTS creates a tunneling gap. The gap sizes extracted from the calculated band structures for BSTS thickness ≤10 quintuple layers (QL), where 1QL≈ 1 nm[19], are shown in Supplementary Figs. 1, 2. The parity of the hybridization gap is evaluated in Supplementary Table 1. A negative parity of the gap occurs when the hole sub-band is at a higher energy level than the electron sub-band, and thus the total parity reads negative, analogous to the description of the negative gap due to the bulk band inversion in 3D TIs. Similar to the binary $Bi_2Se_3$ and $Bi_2Te_3$ 3D TI compounds[2–4], the parity of BSTS hybridization gap exhibits an oscillatory pattern, indicating switching between the band and QSH insulators via thickness modulation as shown at the zero-field limit in Fig. 1a.

The hBSTS exhibits unique electromagnetic responses. In a perpendicular magnetic field, the surface states' hybridization essentially lifts the degeneracy of the $N = 0$ surface Landau level (LL) and causes a

splitting into an electron and hole-like $N = 0$ sublevels[20]. The hybridization gap can thus be defined as the energy difference between the two sublevels, denoted by $E_0$. The analytical formula of $E_0$ reveals a linear relation with magnetic field, expressed as $E_0 = \widetilde{C}_0 + \frac{eB}{\hbar}\widetilde{C}_2 - \frac{\mu_B g B_{21}}{2}$, where $\mu_B$ and $g$ are the Bohr magneton and effective g-factor; $\widetilde{C}_0$ and $\widetilde{C}_2$ are the parameters related to the constant and quadratic terms of the topological surface states' Hamiltonian, and the third term describes the Zeeman energy splitting of surface bands (Fig. 1b). Meanwhile, in the presence of an external perpendicular electric field, the surface states' Hamiltonian is again modified by the electric field as $H_U = e\mathbf{E}_z \cdot \mathbf{z}$[15]. The external electric field can induce a Rashba-type splitting in the surface band[15,16] due to the structure inversion asymmetry in the gapped surface states, resulting in a shrinkage of the hybridization gap (Fig. 1b).

Combining the responses of the hybridization gap to magnetic and electric fields, we construct phase diagrams for hBSTS with trivial and non-trivial topological phases, based on analytical formulations, as illustrated in Fig. 1a. The figures investigate and compare the switching of topological phases in thickness range with distinct topology. For hBSTS with an inverted surface gap, both perpendicular magnetic and electric fields are accountable for the termination of the QSH gap by breaking the time-reversal and inversion symmetries, respectively. The critical switching fields, denoted as $B_c$ and $E_c$, draw the maximum magnetic and electric fields for the QSH state at which boundaries the surface gap vanishes. In contrast, the normal surface hybridization gap reveals noteworthy phase switching only in electric field, while the out-of-plane magnetic field results in monotonic widening of the surface gap[21]. The closure of trivial gap and reopening of inverted gap as controlled by electric field is another route to realize a QSH state.

### Hybridization gap

Experimentally we have employed three different methods to evaluate energy gaps in the hBSTS. The first method is thermal activation

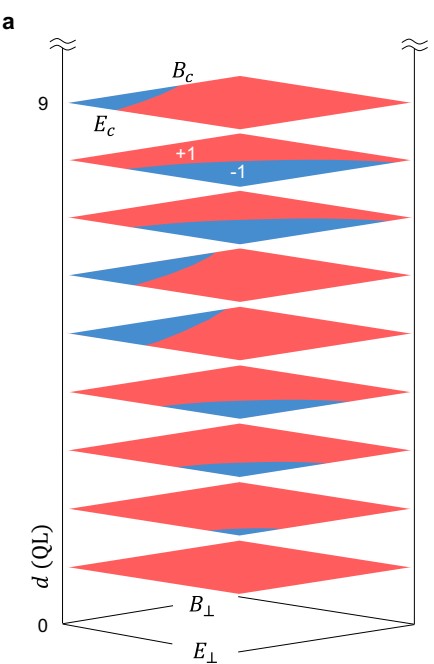

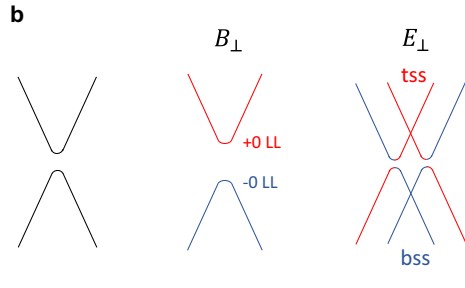

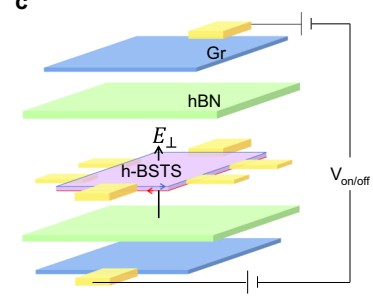

**Fig. 1 | Topological phase transitions. a** Topological phase diagrams for parity of the surface hybridization gap modulated by external magnetic and electric fields acting perpendicular to the c-axis plane of the variable thickness hBSTS. The red and blue color codes represent the opposite sign of parity as a representation of the normal and inverted surface hybridization gaps, respectively. The zero-field gap parity is determined by DFT. The diagrams are drawn based on the Hamiltonian of topological surface states $H_{SOC}$ under distortions of magnetic $H_Z$ and electric $H_U$

fields. **b** Surface band structure of the hBSTS and its evolution under the external magnetic and electric fields. The zeroth LLs bands reside on the surface band edges are labeled, while the higher order LLs are omitted for simplicity. The transverse electric field shifts the top and bottom surface bands and results in the Rashba-like splitting as the illustration. **c** Schematic of an electric field controlled topological transistor device based on an hBSTS in a vdW dual-gating configuration.

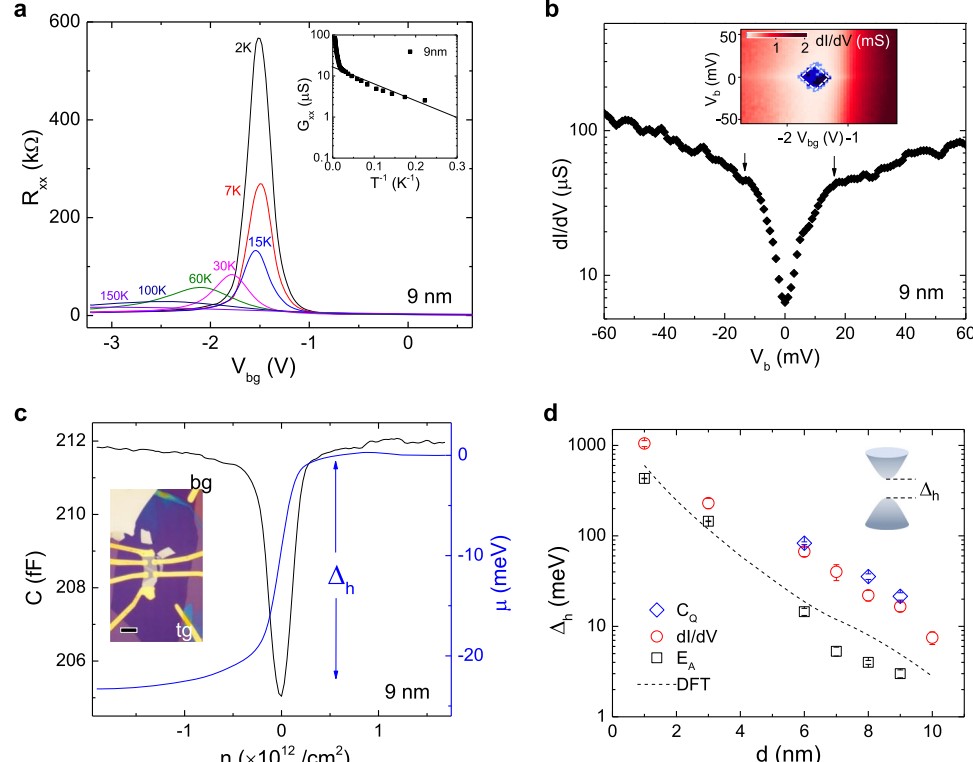

**Fig. 2 | Probing hybridization gap. a** $R_{xx}$ versus $V_{bg}$ plots at different temperatures. Inset in (**a**) is an Arrhenius plot of $G_{xx}$ versus $T^{-1}$ for $V_{bg}$ fixed at the CNP. **b** $dI/dV$ versus $V_b$ curve at insulating region ($V_{bg} \approx -1.5$ V). Inset in (**b**) is a color map of $dI/dV$ as functions of bias voltage ($V_b$) and $V_{bg}$. **c** $C$ and $\mu(n)$ as a function of $n$ induced by $V_{bg}$ with $V_{tg}$ fixed at the overall CNP. The data in (**a**–**c**) were collected on the same 9 nm hBSTS device. Inset in (**c**) is an optical image of an 8 nm hBSTS vdW heterostructure device. Scale bar in (**c**) inset is 10 μm. **d** Hybridization gap ($\Delta_h$) obtained from the three different approaches plotted in log scale as a function of BSTS flake thickness. Error bars in (**d**) are the standard deviation from the fittings. The dashed line in (**d**) is the $\Delta_h$ extracted from our DFT calculations. Inset in (**d**) is a schematic of a hybridized surface band.

energy by fitting the temperature dependent conductance. Figure 2a shows the four-terminal resistance ($R_{xx}$) as a function of back gate voltage ($V_{bg}$) measured at different temperatures for a 9 nm hBSTS. The amplification of $R_{xx}$ peak with the decrease in temperature implies an insulating state developed at charge neutrality point (CNP) due to the intersurface hybridization. The inset of Fig. 2a displays the $G_{xx}$ (=1/$\rho_{xx}$) data taken at the CNP on an Arrhenius plot, where two thermally activated conductance slopes corresponding to the bulk (>100 K) and surface activation (<100 K) gaps are observed. Fitting of the second slope to the activation relation $G_{xx} = G_{xx}^0 \exp(-E_A/2k_BT)$[22] yields a surface activation energy ($E_A$) of ~3 meV. The second method is to probe the non-linear current–voltage characteristics by measuring the differential conductance ($dI/dV$). Figure 2b (inset) displays a $dI/dV$ map as functions of bias voltage ($V_b$) and $V_{bg}$ for the 9 nm hBSTS. As the $V_{bg}$ sweeps across the CNP, the $dI/dV$ reaches an overall minimum, which elucidates the nature of the surface gap. The diamond-shaped feature arises from charge transport across the surface states when the chemical potential is aligned to or detuned from the hybridization gap at the CNP[23]. A tunneling gap of ~16 meV is determined from the $dI/dV$ versus $V_b$ plot (Fig. 2b) across the center minimum. The third approach is to determine the chemical potential relation, $\mu(n)$, by integrating the reciprocal quantum capacitance, $1/C_Q$, with respect to charge density ($n$)[24]. The capacitance dip at the CNP ($n \approx 0/\mathrm{cm}^2$) (Fig. 2c) indicates a minimum density of state (DoS) corresponding to the insulating state. The $\mu(n)$ plotted in Fig. 2c reveals a step feature, where the step height of ~21 meV gives the hybridization gap for the 9 nm hBSTS.

The hybridization gap sizes as a function of hBSTS flake thickness extracted from the three methods are summarized in Fig. 2d. The details of the layer-dependent hybridization gap analyses are presented in Supplementary Fig. 3–6. We note that the thermal activation

is often smeared by disorder potential fluctuation, resulting in a much smaller activation energy gap than the actual surface gap especially when gap size is comparable to the disorder[25]. Despite varying values, these three methods reveal the same trend of nearly exponential decay with the increase of hBSTS thickness. The hybridization gap can thus be approximated by the relation as $\Delta_h \propto e^{-\lambda d}$[26], where the characteristic length ($\lambda$) is fitted to -0.44±0.02 – 0.66±0.01 $\mathrm{nm}^{-1}$ for the three approaches. This exponential dependence confirms the single-particle gap nature by excluding the predicted many body effects, such as topological excitonic states in the overlapped thickness range, as they are less sensitive to the change in film thickness[27]. Furthermore, our DFT calculations (Supplementary Fig. 1) based on the molecular structure of BSTS, as depicted in Fig. 2d, are in good agreement with our experimental trend, further verifying its single-particle origin. We note that our measured single-particle tunneling gap of hBSTS crosses over to the 3D limit at a larger thickness than the previously measured crossover thickness in $Bi_2Se_3$[5]. Such discrepancy can be explained by the resolution limit of ARPES in resolving gap size of the order of meV energy scale. A more detailed triggering of the hybridization gap in $Bi_2Se_3$ using a tight-binding model[28] resolves the exponentially decay of the hybridization gap at thickness beyond 6QL. Sub-meV hybridization gaps were also detected in 12–17 nm $Bi_2Se_3$ using phase-coherent transport[9].

## Normal and inverted gaps

We have noted two inverted surface gap regimes at hBSTS thickness of 9 QL (<10 meV) and 5–6 QL (30–40 meV), based on our DFT calculations of the surface gap topology (Supplementary Table 1). The transport properties in the first inverted regime are studied. Different from the monotonically increasing $R_{xx}$ for the 9 nm hBSTS, the 10 nm

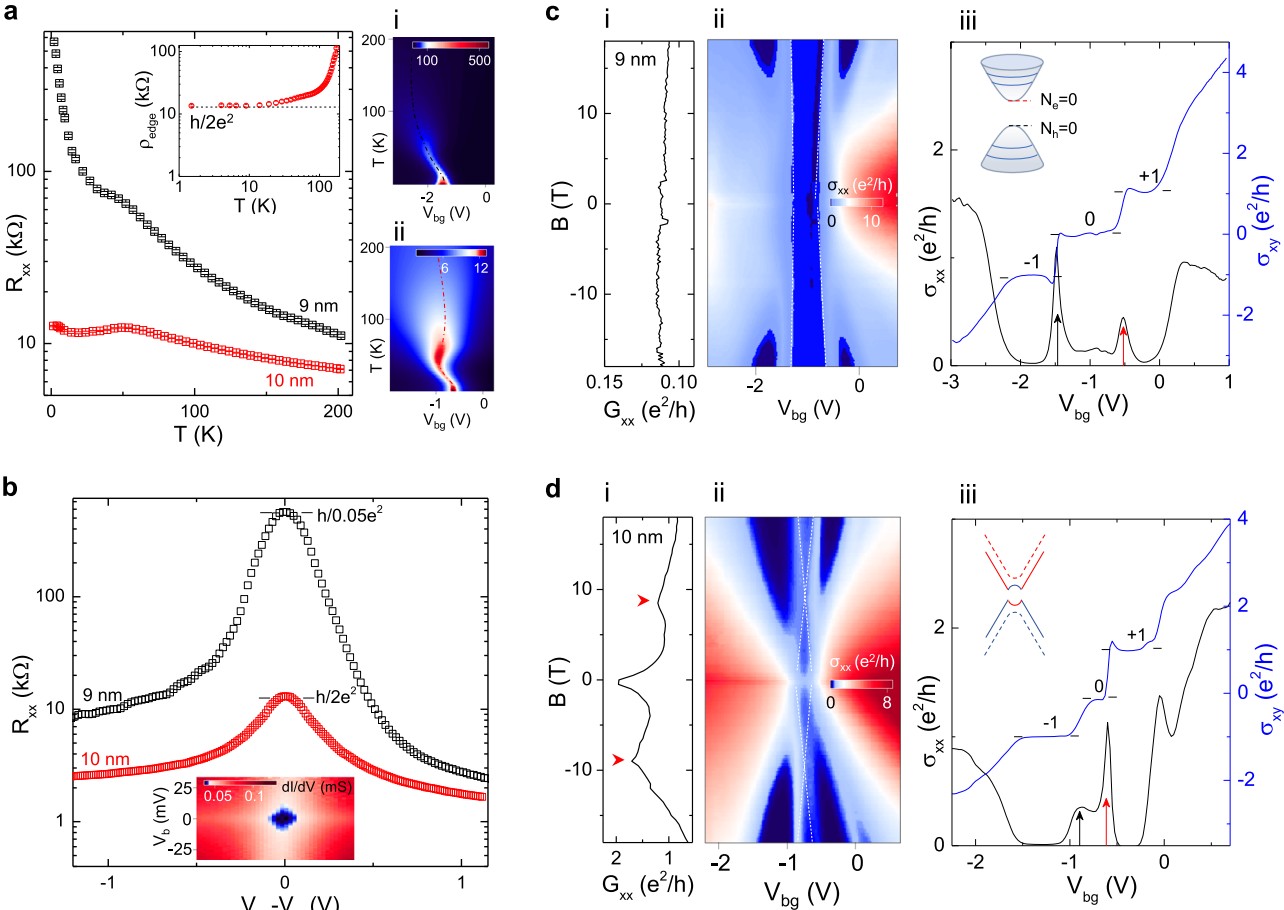

**Fig. 3 | Normal and inverted gaps. a** $R_{xx}$ versus temperature for the 10 nm and 9 nm hBSTS. Error bars in (**a**) are the standard deviation of the extracted $R_{xx}$ values. Inset in (**a**) is $\rho_{edge}$ calculated for the 10 nm hBSTS as a function of temperature. The horizontal dashed line corresponds to h/2e². Right panels in (**a**) are color maps of $R_{xx}$ as functions of $T$ and $V_{bg}$ for the (i) 9 nm and (ii) 10 nm hBSTS. Unit of the $R_{xx}$ is kΩ. Dashed lines in the color maps trace the CNPs at different temperatures. **b** $R_{xx}$ versus $V_{bg}-V_D$ for the 10 nm and 9 nm hBSTS. Inset in (**b**) is a color map of dI/dV as function of $V_b$ and $V_{bg}$ for the 10 nm hBSTS. (i) Plots $\sigma_{xx}$ and $\sigma_{xy}$ versus $V_{bg}$ at the magnetic field of 18 T for the (**c**) 9 nm and (**d**) 10 nm hBSTS. Inset in (**c**) is a schematic of the LLs form in the hybridized surface state. Inset in (**d**) is a schematic of surface band structure evolution under magnetic field for the 10 nm hBSTS with inverted gap. Black and red arrows in line profiles in (**c**) and (**d**) present the two N = 0 LL bands residing at the hole and electron band edges, respectively. (ii) Color maps of the $\sigma_{xx}$ as functions of magnetic field and $V_{bg}$ for the (**c**) 9 nm and (**d**) 10 nm hBSTS. The white dashed lines in color maps in (**c**) and (**d**) trace the development of N = 0 LLs in electron and hole sublevels with magnetic field. (iii) $G_{xx}$ versus $B$ taken at the CNPs for the (**c**) 9 nm and (**d**) 10 nm hBSTS. Red arrows in (**d**) point to the $B_c$ for the transition from QSH to QH states.

sample shows $R_{xx}$ maximum saturating at ~12 kΩ (~h/2e²) at temperature below 50 K. This weak temperature response in the 10 nm hBSTS at low temperature suggests a finite conductance state existing/developing in the hybridization gap[29]. The full data set of $R_{xx}$ at all temperatures for the 9 nm and 10 nm hBSTS as presented in color maps of Fig. 3a(i) and (ii), respectively, further elaborate their distinct temperature dependence behaviors. Figure 3b compares the gate-dependence of $R_{xx}$ at temperature of 1.5 K for the 9 nm and 10 nm hBSTS. The $R_{xx}$ reaches a maximum value of ~500 kΩ (~h/0.05e²) as the chemical potential is tuned into the surface gap, indicating a normal insulating state for the 9 nm hBSTS. Contrary to the strong resistive signal, the 10 nm hBSTS exhibits a finite resistance of ~h/2e² within the hybridization gap regime as indicated by its dI/dV color map inserted in the figure. This again supports the inverted nature of the surface gap for the 10 nm hBSTS. We further show that the two-terminal and non-local resistances measured in different configurations agree with the expected values for helical edge states derived from the Landauer–Buttiker formalism[30] (Supplementary Fig. 7 and Supplementary Table 3). Similar observations are reproduced in an additional 10 nm hBSTS device (Supplementary Fig. 8). However, we note that similar transport features were not observed in the second inverted surface gap regime (5–6 QL) as identified by our DFT calculations. One

possible reason is the significant reduction of the mobility below 7 nm, thus shortening the mean free path and obscuring the transport signature for the 2D topological phase. A more complex device configuration as implemented in monolayer WTe₂[10] will be needed to clarify the QSH phase in the second inverted regime.

To quantitatively evaluate the bulk surface and edge transport, we construct a simple conductance model by considering the total conductance as a parallel sum of the surface and edge conductance, $G_{xx} = G_{surf} + G_{edge} = \frac{1}{\rho_{surf}} + \frac{1}{\rho_{edge}}$. The surface resistivity ($\rho_{surf}$) exhibits a thermally activated behavior with temperature as ~exp($\alpha T$), whereas the edge resistivity ($\rho_{edge}$) will saturate in the quantum limit of ~h/2e² at sufficiently low temperature. Since the edge channel is absent in the normal gap, the total resistivity for 9 nm hBSTS can be simplified to $\rho_{xx}$ (9 nm) = $\rho_{surf}$. Thus, the total resistivity for 10 nm hBSTS can be approximated to the relation as $\frac{1}{\rho_{xx}(10\,nm)} \approx \frac{1}{\rho_{xx}(9\,nm)} + \frac{1}{\rho_{edge}}$. To verify this relation, we plot the $\rho_{edge}$ as a function of temperature as inserted in Fig. 3a. Figure 3a shows excellent quantization of the edge conductance at a value of h/2e² in accordance with the quantum spin Hall state. This analysis also shows that the quantization of $\rho_{edge}$ persists up to 20 K, indicating a QSH gap of ~2 meV for the 10 nm hBSTS. We note that this QSH gap is smaller than the surface hybridization gap

(~8 meV) as estimated from the differential conductance measurement. Given that the device size is larger than the inelastic mean free path, backscattering between the helical edge states[31] can occur at higher temperature, which can cause the deviation of $\rho_{edge}$ from quantization.

## Magnetic-field response

The normal and inverted surface gaps of hBSTS present distinct behaviors in perpendicular magnetic field. We first discuss the normal gap feature. Line profiles in Fig. 3c plot the longitudinal conductivity ($\sigma_{xx}$) and Hall conductivity ($\sigma_{xy}$) as a function of $V_{bg}$ measured at a magnetic field of 18 T for the 9 nm hBSTS. The two N = 0 LL bands developed in the $\sigma_{xx}$ plot are assigned to the electron and hole band edges of the hybridization gap as illustrated by the schematic of the surface band diagram in the inset of Fig. 3c. This splitting of N = 0 LLs is a key signature of the Landau quantization of intersurface hybridization[20,21]. The established $\nu$ = 0 plateau in the $\sigma_{xy}$ plot within the N = 0 LL, together with the development of $\nu$ = −1 and +1 QH plateaus symmetrically about zeroth plateau, further supports the LLs of the hybridized surface states. Color map of $\sigma_{xx}$ in Fig. 3c illustrates the development of the N = 0 LLs as a function of magnetic field. Similar behaviors were also observed for the 8 nm hBSTS (Supplementary Fig. 9). The N = 0 LL energy spacing ($E_0$) derived from the $\mu(n)$ relation shows a linear magnetic field dependence (Supplementary Fig. 10), in agreement with the analytical model[21]. The fitting of $E_0$ to magnetic field yields $\tilde{C}_0$ ~ 0.016 eV and $\tilde{C}_2$ ~ 80 eV Å$^2$ for the 9 nm hBSTS, which is comparable to the fitting parameters of the surface states' Hamiltonian[32].

The gate-dependent $\sigma_{xx}$ and $\sigma_{xy}$ plots for the inverted gap hBSTS (10 nm) measured at the magnetic field of 18 T (Fig. 3d) show a similar type of N = 0 LLs splitting. The relatively narrow zeroth LL plateau width in charge density compared to the 9 nm hBSTS is consistent with its smaller size of hybridization gap as previously discussed in the gap analyses. Interestingly, the development of the N = 0 LLs with magnetic field traced by the dashed lines along the $\sigma_{xx}$ minimum in Fig. 3d reveals an intriguing feature. The two N = 0 sublevels develop oppositely as revealed by the $\sigma_{xx}$ color map at low magnetic field, signifying the inversion of electron and hole sublevels at zero field. As the magnetic field increases, these two sublevels eventually cross and develop into normal QH states with the two sublevels interchanged. This is equivalent to an inverted-to-normal surface gap evolution with magnetic field to transform from its topologically non-trivial into a trivial gap state as illustrated in surface band diagram in Fig. 3d. This magnetic field mediated TPT is similar to the band crossing for the QSH effect observed in HgTe/CdTe quantum well 2D TIs[33]. The sublevels' crossing feature forming symmetrically about the opposite magnetic field further confirms this magnetic field-driven topological transition.

Figure 3c, d(i) compare $G_{xx}$ curves as a function of magnetic field for the 9 and 10 nm hBSTS, respectively, with their chemical potentials tuned into surface gaps. The levels crossing point at $B_c$ of ~9(−9) T is clearly resolved in the $G_{xx}$ curves of the 10 nm hBSTS (Fig. 3d). This $B_c$ appears to be consistent with its surface gap size (~8 meV) obtained from gap analyses and the opening rate of ~1 meV/T revealed in 9 and 8 nm hBSTS (Supplementary Fig. 10). A similar feature is captured in a parallel magnetic field (Supplementary Figs. 11, 12) except no QH state evolves. Comparing with quantum wells, the QSH state in hBSTS persists strongly in magnetic field. The $G_{xx}$ falls off more slowly with the magnetic field as inferred by the width of $G_{xx}$ peak centered at zero magnetic field of ~2 T (~28 mT for HgTe/CdTe quantum wells[31]). Also, the slope of $dG_{xx}/dB$ in linear region ($B$ = 0–2 T) of ~0.2–0.3 $e^2/h/T$ is nearly two orders of magnitude smaller than the quantum wells[31,34]. The $dG_{xx}/dB$ slope reflects directly the disorder strength ($W$) of the QSH state[35]. In our case, the estimated $W$ of a few meV indicates a moderate to weak disorder ($W \lesssim \Delta_h$) regime with insignificant bulk scattering, thus providing a cleaner platform to study the one-dimensional edge channel.

## Electric-field response

An external electric field can also modulate the surface bands to induce TPTs. The effect of a transverse electric field can be triggered by controlling voltages on top and bottom gates through the hBSTS as illustrated in Fig. 1c. The diagonal resistance maximum along the transverse electric field observed in the dual-gate mapping (Supplementary Fig. 15) presents a key feature resulting from the strong coupling between surface states. To better elucidate the electric field response, we converted the dual-gate voltages into the displacement field ($D$) versus total charge density ($n$) relation[36], as presented in descending order of thickness in Fig. 4a–d. Magnitudes of the corresponding hybridization gaps at zero $D$ are revealed by their d$I$/d$V$ maps inserted in the figures. Line profile of $\rho_{xx}$ versus $D$ (Fig. 4a) for the inverted gap hBSTS (10 nm) shows a gradual decrease from its quantized $h/2e^2$ value at large $D$, inferred as a transition into a semi-metallic phase. Whereas for the normal gap hBSTS (9 nm), the $\rho_{xx}$ at the CNP is suppressed by more than one order of magnitude at $D$ ~100 mV/nm and tends to saturate to a value of order $h/e^2$ at large $D$, suggesting a significant gap reduction by $D$ (Fig. 4b). This $\rho_{xx}$ change is highly symmetric with respect to opposite polarities of $D$. The suppression of $\rho_{xx}$ in $D$ becomes less pronounced in thinner hBSTS (Fig. 4c, d), further signifying modulation of hybridization gap with $D$. For comparison, we plot the change in resistivity $\Delta\rho_{xx}/\rho_{xx}(0)$ for the variable thickness hBSTS in the normal gap regime in Fig. 4e. Consistently, the $\Delta\rho_{xx}/\rho_{xx}(0)$–$D$ response scales down monotonically with decreasing thickness of hBSTS.

To further investigate the $\Delta_h$–$D$ relation, we again deploy our three gap estimation methods to probe the gap size for the 9 nm hBSTS versus $D$, as summarized in Fig. 4f. The gap analyses are examined in detail in Supplementary Fig. 16–19. Consistent with the $\rho_{xx}$–$D$ trend, a significant reduction in hybridization gap with $D$ is indicated by all three methods. The same trend is observed in opposite polarity of $D$, suggesting that the gap reduction is due to broken structure inversion symmetry between top and bottom surface states by transverse electric field[15,16]. The appreciable reduction of the surface gap is attributed to the electric field that is effectively applied through the vdW layers. The gap size reaches a minimum saturation value of <1 meV, indicating a critical displacement field, $D_C$ at ~250 mV/nm. Similar analyses are implemented for the 8 and 6 nm hBSTS, while for simplicity only gap calculated from $C_Q$ are included in Fig. 4f. Their $D_C$ can be extrapolated from the curves as ~400 and ~750 mV/nm for the 8 and 6 nm hBSTS, respectively. Meanwhile, an effective gap-closing strength of ~1.2 eÅ can be estimated from the $\Delta_h/D$ ratio for the 9 nm hBSTS, which is comparable to DFT calculated value $\Delta_h/E$ of ~2–3 eÅ (Supplementary Figs. 13, 14).

A direct consequence of the surface gap termination is the emergence of topological edge state due to the change in surface band topology. To analyze this effect, we again apply the parallel conductance addition model and evaluate $\rho_{edge}$ for the 9 nm hBSTS at applied $D$ values. Assuming an edge channel evolves with $D$, $\rho_{xx}(D)$ will follow the relation: $\frac{1}{\rho_{xx}(D)} = \frac{1}{\rho_{xx}(0)} + \frac{1}{\rho_{edge}}$. From this relation, $\rho_{edge}(D)$ is obtained at different temperatures and plotted in Fig. 4g showing suppression of thermal activation with the increase in $D$. This analysis shows that, at large $D$ (250 mV/nm), $\rho_{edge}$ behavior resembles the temperature curve of the inverted gap hBSTS (inset of Fig. 3a) and approaches the $h/2e^2$ value at low temperature (<40 K), indicating an emerging helical edge state. This gap-closing mechanism mediated by transverse electric field is illustrated by the evolution of surface spectra (insets of Fig. 4g), which offers an alternative route toward high-temperature topological edge channels.

In summary, we realized distinct topological phases by mapping the evolution of the surface gap of BSTS 3D TI with thickness, temperature, and transverse magnetic and electric fields in the quantum tunneling regime. The trivial and topological phases in hBSTS were identified by their diverging and finite (~$h/2e^2$) resistances,

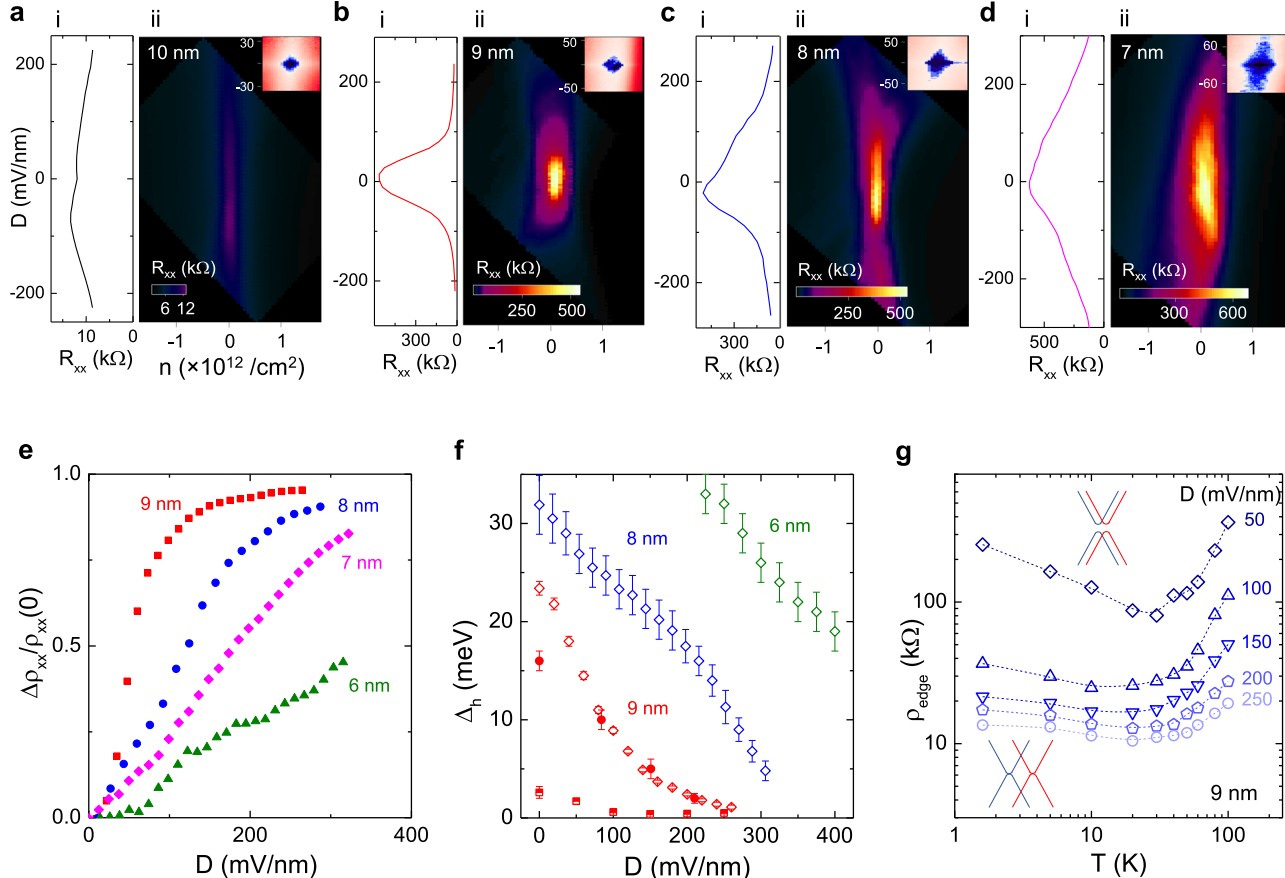

**Fig. 4 | Surface gap inversion by external electric field.** (i) Color maps of $R_{xx}$ as functions of displacement field ($D$) and total charge density ($n$) for the different thickness hBSTS in the order of (**a**) 10 nm, (**b**) 9 nm, (**c**) 8 nm, and (**d**) 7 nm. The $D$ and $n$ are calculated from dual-gate voltages as: $D = \frac{1}{\varepsilon_o}\left(\frac{\varepsilon_b \Delta V_{bg}}{d_b} - \frac{\varepsilon_t \Delta V_{tg}}{d_t}\right)$, and $n = C_{bg}\Delta V_{bg} + C_{tg}\Delta V_{tg}$. (ii) Line profiles of $R_{xx}$ versus $D$ for the respective thickness taken at $n \approx 0$ cm$^{-2}$. Inset in (**a**–**d**) is the d$I$/d$V$ color maps as functions of $V_b$ and $V_{bg}$.

for the respective samples. **e** $\Delta\rho_{xx}/\rho_{xx}(0)$ as a function of $D$ for different thicknesses hBSTS. **f** Hybridization gaps as a function of $D$ for the variable thickness hBSTS. The gap size determined from the three different methods is labeled with different symbols as $C_Q$ (rhombus), d$I$/d$V$ (square), and $E_A$ (circle). Error bars in (**f**) are the standard deviation from the fittings. **g** $\rho_{edge} = 1/[\rho_{xx}(D)^{-1} - \rho_{xx}(0)^{-1}]$ as a function of temperature for the 9 nm hBSTS at different D. Inset in (**g**) is a schematic of surface band structure evolution under electric fields for the 9 nm hBSTS.

respectively, in response to temperature and gate voltages. The helical edge state reveals more than one order of magnitude smaller disorder strength compared to HgTe/CdTe quantum wells. By studying the development of zeroth LL sub-bands in the perpendicular magnetic field, we observed an inverted-to-normal surface gap crossing accompanied by a transition from QSH to QH edge states. We further realized surface gap closing with electric field, together with an emerging edge conductance for the normal insulating gap hBSTS, implying a transition between trivial and topological phases mediated by the electric field. These compelling signatures of TPTs and the reversible switching mechanisms hold promise for TI-based topological field-effect transistors (Supplementary Table 4).

## Methods
### DFT calculations
Our calculations were performed within the framework of density functional theory as implemented in the Vienna ab initio simulation (VASP)[37]. The projector augmented wave potentials were adopted with the generalized gradient approximations of Perdew–Burke–Ernzerhof exchange-correlation functional, and the cutoff energy was set to 520 eV. The relaxation is performed until all forces on the free ions converge to 0.01 eV/Å and the Monkhorst-Pack k-point meshes of 10 × 10 × 1 were used, which have been tested to be well convergence. The vacuum space is at least 20 Å, which is large enough to avoid the interaction between periodical images. The different BSTS systems

were treated with virtual crystal approximation. The van der Waals interaction is described by DFT-D3 method. In addition, the spin-orbit-coupling was included in the calculations of the electronic structure.

### Device fabrication
Variable thicknesses of hBSTS crystal flakes were exfoliated from the bulk crystal[38] and then transferred using a micromanipulator transfer stage into the heterostructures of Gr/hBN sandwiched layers. We fabricated the hBSTS devices into the Hall bar configuration with Cr/Au (2 nm/60 nm) as the contact electrodes. The top and bottom Gr/hBN layers serve as the gate-electrode/dielectric layers for applying of perpendicular electric field to the hBSTS (Fig. 1c). The thickness of the BSTS flakes was measured by a Bruker Dimension Icon atomic force microscopy. The device dimensions were obtained from the images taken by a high-resolution optical microscope. hBSTS devices with thickness ranging from 10 nm down to 1 nm were fabricated and studied. The device specifications are presented in Supplementary Table 2.

### Measurements
Low-temperature electrical transport measurements were performed in a helium-4 variable temperature insert with a base temperature of 1.6 K and magnetic field up to 9 T. Four probe resistances of the devices were measured using a Stanford Research SR830 lock-in amplifiers operated at a frequency of 17.777 Hz. The devices were

typically sourced with a constant AC excitation current of 10–20 nA. Two Keithley 2400 source meters were utilized to source DC gate voltages separately to the top and bottom Gr gate electrodes. Variable temperature transport measurements were controlled by a Lakeshore temperature controller. The differential conductance was measured using a Stanford Research SR830 lock-in amplifier coupled to a model 1211 current preamplifier. The devices were sourced with an AC excitation voltage of 100 μV. The DC bias voltage in a range from 40–1200 mV was swept across the source-drain electrodes. Magneto-transport measurements at high magnetic field were carried out in a helium-3 variable temperature insert at a base temperature of 0.3 Kelvin and magnetic field up to 18 tesla based at the National High Magnetic Field laboratory. Two synchronized Stanford Research SR830 lock-in amplifiers were used to measure the longitudinal and Hall resistances concurrently on the BSTS devices. Capacitance was measured in a capacitance bridge configuration[24] connected between the BSTS device and the parallel gold strip as a reference capacitor. Two synchronized (at a frequency of ~50–70 kHz) and nearly equal-amplitude AC excitation voltages (range of 15–40 mV) were applied separately to the top and bottom Gr gates, whose relative magnitude was chosen to match the ratio of geometric capacitances of top and bottom surfaces. A third AC excitation voltage was applied to the reference capacitor with the amplitude set to null the measured signal. The reference capacitors were calibrated to be ~300–400 fF using a standard capacitor (Johanson Technology R14S, 1 pF). The capacitance data were acquired by monitoring the off-balance current at the balance point as the DC gate voltages were changed.

## Data availability
The data that support the findings of this study are available within the article and its supplementary information files. Source data are provided with this paper.

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

## Acknowledgements

This material is based upon work supported by the National Science Foundation, the Quantum Leap Big Idea under Grant No. 1936383. A portion of this work was performed at the National High Magnetic Field Laboratory, which is supported by National Science Foundation Cooperative Agreement No. DMR-1644779 and the State of Florida. The authors acknowledge Brian Skinner for helpful comments.

## Author contributions

S.K.C. and V.V.D. designed, conducted the experiments, and prepared the manuscript. K.W. and T.T. provided single crystal hexagonal boron nitride. T.D.S. provided a single crystal BiSbTeSe$_2$ three-dimensional topological insulator. L.L. and F.L. performed theoretical calculations to support the experimental data. All authors contributed to the discussion of the results and approved the final version of the manuscript.

## Competing interests

The authors declare no competing interests.
