## [Peer Review File · Nature Communications]

Emergent Helical Edge States in a Hybridized Three-Dimensional Topological InsulatorREVIEWER COMMENTS

Reviewer #1 (Remarks to the Author):

This work is addressing a longstanding issue on observing transport signatures of hybridization between topological surface states and the resulting two-dimensional quantum spin Hall or quantum Hall (in the presence of B-field) phases. By changing the layer thickness, they observe the alternating trivial-nontrivial behavior as theoretically suggested previously. Also, they also observe E-field-driven inversion of hybridization gap, which was also suggested previously but has not been observed in experiment up to my knowledge.

Although most of theoretical predictions on topological insulators, including the main subject of this work, have been known for a long time, but their experimental verification has remained challenging. This work tackles such difficult task and presents here beautiful data and clear presentation of the results. Hence I am happy to recommend this manuscript in Nature Communications.

Before resubmission, I'd like the authors to define some symbols, like V_b and V_{bg} , for some readers new to this field. Also, in the inset of Fig. 3a, I think it is not mentioned either in the caption or main text that the horizontal based line corresponds to $h/2e^2$.

Reviewer #2 (Remarks to the Author):

The manuscript by Chong et al reports the transport measurement on reduced-thickness 3D topological insulator BiSbTeSe₂, with DFT calculation. A variety of devices with different thickness were measured. A gapped to gapless transition were observed when the device thickness is increased from 9 nm to 10 nm. For thickness within 9 nm, a monotonically increasing gap were observed with decreasing the thickness, consistent with the calculations. Interestingly, the 10-nm device shows near quantized conductance at low temperature, and the conductance shows a transition with increasing the perpendicular field. This is in strong contrast with the 9-nm device, which shows a clear activated gap, and was maintained under perpendicular magnetic field up to 18 T. Furthermore, a displacement field induced gap closing with the resistance decreasing toward $h/2e^2$ is observed in the 9 nm, showing a displacement field induced gapping closing. The manuscript is well-written, covering broad background introduction, systematic study of the thickness, temperature, magnetic field, doping density, displacement fields dependence, and detailed discussions on the experimental results. The data is technical sound for me, and notably, the systematic study of the thickness dependence with the observation of gapped to gapless transition is very impressive.

I could recommend for publication on Nature communication if the following questions are well addressed.

1. The key observation was shown in a 10nm device, where only one device is shown in proving this. Does the author have performed measurements in a thicker device? Say 11 nm? Moreover, can the authors plot the Fig. 3b in separate linear-scale plots? I wonder how much the quantized it is closed to $h/2e^2$.
2. It would be interesting if the authors could show a fix V_{bg} line cut of σ_{xx} and σ_{xy} slightly away from the CNP. The data shows asymmetry to perpendicular field, is there any (quantum) anomalous Hall effect?
3. For the displacement field dependent data shown in Fig. 4e-g, can the dual gates be inverted? Why there are no data for $D < 0$? Also, why the data in Fig. a-d symmetric to $D=0$?
4. Dose the authors have the displacement field dependent data for 10 nm devices as shown in Fig. 4? Is it gap tunable?
5. I feel very terrible when reading through the manuscript, not for the data quality, but for the figure and caption illustration. I urgent the authors to carefully check the figures, including all notions. Key parameters should be added to the figures. For example:
 - a. Fig. 2, device thickness should be added.
 - b. For example, “)” is missing in Fig. 2b inset.

- c. Fig. 3c & d should label the thickness.
- d. Fig. S3, thickness should be added.
- e. Fig. S5, the thickness & gap sized should be noted in the S5a-g.
- f. More need be carefully checked by the authors.

Other comments/questions that might be helpful:

1. The first observation of the quantum anomalous Hall effect is somehow missing when introducing the magnetic 3D TIs.
2. The notions of the physical parameters should be illustrated. For example, V_{bg} ?
3. Adding subtitles to summary different topics/titles might be helpful.
4. Why the gap is characterized by $\exp(-\Delta/k_B T)$, not $\exp(-\Delta/2k_B T)$?
5. In supplementary information, Table S3, the Strained InAs/InGaSb is categorized to 3D TI. Why? If the authors read Phys. Rev. Lett. 119, 056803 (2017), it should be helpful.

Reviewer #3 (Remarks to the Author):

Chong et al fabricated a few devices based on the exfoliated BiSbTeSe₂(BSTS) flakes and performed systematic transport measurements. The authors used three approaches to acquire the size of the hybridization gap in these devices. By combining the DFT calculations, the authors claimed the observation of the quantum spin Hall effect in a 10QL device. The story is interesting and the paper is written well. However, I have two major concerns about the claim of this paper.

(1) As described on Page 3 and shown in Fig. 1a, the theoretical calculations clearly show that the BSTS hybridization gap exhibits an oscillatory pattern between the QSH and normal insulators. In other words, the quantum spin Hall phase appears in the devices within a broad thickness range. However, why the quantum spin Hall phase appears only in the 10QL but disappears in the devices with other thicknesses (e.g. 8 and 9 QL)? I understand that the electric field can tune the devices with different thicknesses into the QSH phase regime easily, as shown in Fig. 4.

(2) How many 10QL devices show the quantum spin Hall effect? If there is only a 10QL device showing the QSH phase. The longitudinal resistance close to $h/2e^2$ might be observed by chance.

Below are my minor concerns:

(3) The 2D-3D TI crossover thicknesses are 6 QL, 4 QL, and 2 QL for Bi₂Se₃, Sb₂Te₃, and Bi₂Te₃, respectively (Ref. 7; PRL 108, 016401(2012); Adv. Mater. 22, 4002(2010)). This crossover thickness should be independent of the electrical properties of the TI films, so I don't quite understand why the 2D to 3D TI crossover thickness of BSTS in this work is much larger. It would be great that the authors should list the parameters used in their DFT calculations and add more discussions about this issue in the paper.

(4) In Fig.3b, the vertical axis is in the log scale, please change it to the linear scale because the log scale will hide the r_{xx} quantization of the QSH phase. Based on the data shown in Fig.4a, I speculate that r_{xx} is larger than $h/2e^2$ but there is no plateau feature by tuning the gate.

(5) The estimated inverted gap for the 10 QL BSTS device is ~ 20 meV. This gap size corresponds to the critical temperature of the QSH phase ~ 200 K, which is much higher than the ~ 40 K reported in this paper. The authors need to add more discussions about this issue in the paper.

(6) The caption of Fig.2 should include the device information.

Based on my above comments, I will not recommend this paper for publication in Nature Communications. If the authors could reproduce this phenomenon in more devices, I will reconsider this work in the future.

Reviewer #4 (Remarks to the Author):

Dear Editor,

after reviewing the manuscript "Emergent Helical Edge States in a Hybridized Three-Dimensional

Topological Insulator", I remain unconvinced that the paper supports the conclusions of any new physics. In addition, I find the manuscript to be quite poorly written, and very hard to follow.

This impression starts in the introduction: For example, the choice of Reference 2 and 3, which are neither the firsts, nor the most well-known papers for observations of magnetic TI's suggest a complete unfamiliarity with the field, as does the fact that the authors appear to be unaware that both 2D and 3D magnetic TIs have been studied, and indeed the transition between the two is discussed in the literature. This of course can be repaired with a proper rewriting of the introduction.

The significant issue however is that the scientific evidence as presented is simply not compelling.

The main claim of the paper relies on observing the helical edge channels under certain conditions (either intrinsically in a 10nm thick sample, or by applying large electric fields to a 9nm thick sample). This claim is based on R_{xx} having a value close to $h/2e^2$, but that could be merely coincidental. To convincingly show this results from edge states, one should use various non-local measurement configurations and show with proper quantized plateau values consistent with the Landauer-Buttiker formalism for two helical edge channels in each configuration. This can easily be done on their existing samples, as they have a Hall bar geometry, which offers multiple possible configurations with distinct expected plateau values.

In addition to this main criticism, here is a list (perhaps incomplete, as the paper is very hard to follow) of additional significant issues which need addressing:

- There is a discrepancy between the gap size obtained from the Arrhenius fit, and the other two methods (Fig. 2d), by about a factor of 2-3. Why?
- What justifies the use of the Arrhenius fit, over for example the variable range hopping. It is clear in the inset to Fig. 2a that the G_{xx} dependence is not linear for a full range of $1/T$. Indeed, with the data for Fig. 2a covering less than a decade, any attempt to extract exponential/power law behaviour is fraught with danger.
- The assignment of the arrows in Fig. 2b (and for the other samples in Fig.S5) seems rather arbitrary, and this affects the extracted gap using this method.
- How can they be sure of the quality of their contacts. The argument (found in the supplement) that they check them by additionally measuring at different gate voltages where the sample is conducting is not convincing, as the contact quality can also vary significantly with gate voltage (especially when the sample approaches its insulating gap.)
- It is quite hard to tell for certain how Fig. 3d (the color map) would look like without the dashed lines, but I suspect they are misleading and trying to guide the readers eye to see something that is not really visible in the data itself.
- The G_{xx} magnetic field sweep in Fig. 3c does not look numerically processed or like a poorly resolved measurement. Moreover, they say in "methods" that they current bias the sample at 10-20 nA and AC 17.777 Hz. Doing current biased measurements on an insulator is not proper and can lead to measurement artefacts. A better measurement of this G_{xx} signal might reveal similar features to the ones observed in Fig. 3d marked with red arrows (that are interpreted as band inversion, or destruction of helical edge states). This should be checked.
- The 10nm sample that has $\sim h/2e^2$ R_{xx} resistance appears to be a single device. All thinner samples are insulating. Is the result reproducible?
- In addition, there is one sample in the supplement labelled as "10nm (no gap)". It is unclear that this means. Is there so much sample to sample variations? How robust are then any of the observations?
- How can it be excluded that the large electric field are affecting the bulk, and activating bulk carriers?
- In Fig. 4g where by applying large electric fields the authors claim the transition to the helical edge channel regime, the ρ_{edge} does not look like it will saturate with electric field strength, but looks like it would drop even further with larger field. This would contradict the entire interpretation. In this picture, one could simply adjust the maximum field to produce a different value than the observed $\sim h/2e^2$ is one wished.
- There appears to be some confusion between ρ_{xx} and R_{xx} in Fig. 4a-d (between the labels on the axis and the caption).
- The manuscript also contains unclear statements like "cleaner surface bands". This reader has no

idea what that should mean?

In summary, the current data does not provide any compelling evidence of the authors claim, and until that is resolved, I cannot recommend this paper for publication.

Responses to Reviewers' comments:

We thank the reviewers for their very constructive comments and suggestions, which we have taken very seriously. We have taken great efforts to fabricate additional devices and take additional data as requested by the reviewers. Because the first author of this paper had graduated and moved on to a postdoctoral position, it took longer to perform the additional work. We apologize for the resulting delay in submitting this revision.

Reviewer #1 (Remarks to the Author):

This work is addressing a longstanding issue on observing transport signatures of hybridization between topological surface states and the resulting two-dimensional quantum spin Hall or quantum Hall (in the presence of B-field) phases. By changing the layer thickness, they observe the alternating trivial-nontrivial behavior as theoretically suggested previously. Also, they also observe E-field-driven inversion of hybridization gap, which was also suggested previously but has not been observed in experiment up to my knowledge.

Although most of theoretical predictions on topological insulators, including the main subject of this work, have been known for a long time, but their experimental verification has remained challenging. This work tackles such difficult task and presents here beautiful data and clear presentation of the results. Hence I am happy to recommend this manuscript in Nature Communications.

Response: We thank the reviewer for appreciating this work.

Before resubmission, I'd like the authors to define some symbols, like V_b and V_{bg} , for some readers new to this field. Also, in the inset of Fig. 3a, I think it is not mentioned either in the caption or main text that the horizontal based line corresponds to $h/2e^2$.

Response: We defined the symbols and added the description of the horizontal based line in the inset of Fig. 3a in the revised manuscript.

Reviewer #2 (Remarks to the Author):

The manuscript by Chong et al reports the transport measurement on reduced-thickness 3D topological insulator BiSbTeSe₂, with DFT calculation. A variety of devices with different thickness were measured. A gapped to gapless transition were observed when the device thickness is increased from 9 nm to 10 nm. For thickness within 9 nm, a monotonically increasing gap were observed with decreasing the thickness, consistent with the calculations. Interestingly, the 10-nm device shows near quantized conductance at low temperature, and the conductance shows a transition with increasing the perpendicular field. This is in strong contrast with the 9-nm device, which shows a clear activated gap, and was maintained under perpendicular magnetic field up to 18 T. Furthermore, a displacement field induced gap closing with the resistance decreasing toward $h/2e^2$ is observed in the 9 nm, showing a displacement field induced gapping closing.

The manuscript is well-written, covering broad background introduction, systematic study of the thickness, temperature, magnetic field, doping density, displacement fields dependence, and detailed discussions on the experimental results. The data is technical sound for me, and notably, the systematic study of the thickness dependence with the observation of gapped to gapless transition is very impressive.

Response: We thank the reviewer for appreciating this work.

I could recommend for publication on Nature communication if the following questions are well addressed.

1. The key observation was shown in a 10nm device, where only one device is shown in proving this. Does the author have performed measurements in a thicker device? Say 11 nm? Moreover, can the authors plot the Fig. 3b in separate linear-scale plots? I wonder how much the quantized it is closed to $h/2e^2$.

Response: In response to the reviewer's comment, we have taken and added transport data for a 12 nm BSTS device for comparison with the thinner devices in S.I. Fig. S3 in the revised manuscript. The temperature dependent data for the 12 nm BSTS is a representative sample for the regime of thicker BSTS (no hybridization gap). In the thicker devices, the R_{xx} peak value will first increase due to the suppression of the bulk conduction, and then gradually decreases with temperature below 60K due to the gapless conducting surface states.

Also, we plotted the Fig. 3b in separate linear-scale plot in S.I. Fig. S7 together with the non-local transport data. As shown in the figure, by controlling the gate voltage to the hybridization gap, the R_{xx} peaks very close to the value of $h/2e^2$.

2. It would be interesting if the authors could show a fix V_{bg} line cut of σ_{xx} and σ_{xy} slightly away from the CNP. The data shows asymmetry to perpendicular field, is there any (quantum) anomalous Hall effect?

Response: We thank the reviewer for mentioning the possibly interesting features in perpendicular magnetic field. As there is no magnetic doping in our crystals, we believe there should not be a (quantum) anomalous Hall effect in our magnetic field data. We show here the line cuts of σ_{xx} and σ_{xy} slightly away from the CNP for both the hole and electron carriers, which show quantum Hall effect at high magnetic field with the suppression of σ_{xx} and the σ_{xy} quantized to h/e^2 .

3. For the displacement field dependent data shown in Fig. 4e-g, can the dual gates be inverted? Why there are no data for $D < 0$? Also, why the data in Fig. a-d symmetric to $D=0$?

Response: We show here the displacement field dependent data for $D < 0$ for the 9 nm and 8 nm hBSTS. Basically, the data for $D < 0$ follows similar trend as the $D > 0$. The data symmetric to $D = 0$ is also consistent with DFT calculations. To answer the reviewer's question regarding symmetry about $D = 0$, we clarify that the electric field will cause a vertical shift of the top and bottom surface bands resulting in a Rashba-like band splitting in hBSTS. This causes a surface gap closing. Applying an opposite electric field will shift the top and bottom surface bands in the opposite direction without affecting the gap closing mechanism. This is due to the overall inversion-symmetric nature of our material and geometry. This situation is quite different, for example, from the electric field induced topological phase transition in InAs/GaSb quantum well [*Phys. Rev. Lett.* **115**, 036803 (2015)]. We have also made this clarification in the revised manuscript.

4. Does the authors have the displacement field dependent data for 10 nm devices as shown in Fig. 4? Is it gap tunable?

Response: Based on the transport data shown in Fig. 4a for the 10 nm device, a large displacement field seems to turn the sample into metallic phase as indicated by the deviation in R_{xx} value from $h/2e^2$. If the reviewer is asking for data similar to Fig. 4f, since this data came from our quantum capacitance measurement, unfortunately we were not able to successfully measure it in the 10 nm device, presumably due to the smaller flake size (parasitic capacitance dominating the measurement) for the 10 nm hBSTS.

5. I feel very terrible when reading through the manuscript, not for the data quality, but for the figure and caption illustration. I urgent the authors to carefully check the figures, including all notions. Key parameters should be added to the figures. For example:

- a. Fig. 2, device thickness should be added.
- b. For example, “)” is missing in Fig. 2b inset.
- c. Fig. 3c & d should label the thickness.
- d. Fig. S3, thickness should be added.
- e. Fig. S5, the thickness & gap sized should be noted in the S5a-g.
- f. More need be carefully checked by the authors.

Response: We apologize for the omissions in the figure and caption illustration. We have corrected the figures and added the appropriate notions and key parameters to the figures in the revised manuscript. The manuscript has been carefully checked by the authors.

Other comments/questions that might be helpful:

1. The first observation of the quantum anomalous Hall effect is somehow missing when introducing the magnetic 3D TIs.

Response: We decided to remove the reference to magnetic 3D TIs as it is not related to the subject of the manuscript. The only reason we had included it previously was to make a comparison of the less well-studied hybridization gap with the exchange gap. Since magnetic TIs are extraneous to the current study, we decided to remove the reference from our manuscript.

2. The notions of the physical parameters should be illustrated. For example, V_b ? V_{bg} ?

Response: We added the notions of the physical parameters, V_b and V_{bg} in the revised manuscript.

3. Adding subtitles to summary different topics/titles might be helpful.

Response: We added subtitles to summarize different topics in the revised manuscript.

4. Why the gap is characterized by $\exp(-\Delta/k_B T)$, not $\exp(-\Delta/2k_B T)$?

Response: We have no preference for one or the other. We have changed the Arrhenius equation for activation gap to $\exp(-\Delta/2k_B T)$ and recalculated the activation gaps for the different thickness BSTS in the revised manuscript.

5. In supplementary information, Table S3, the Strained InAs/InGaSb is categorized to 3D TI. Why? If the authors read Phys. Rev. Lett. 119, 056803 (2017), it should be helpful.

Response: We thank the reviewer for pointing out this mistake and introducing the appropriate reference for the strained InAs/GaInSb quantum well to us. We have now listed the correct category for the strained InAs/GaInSb and cited the reference in the revised manuscript. We apologize for our error.

Reviewer #3 (Remarks to the Author):

Chong et al fabricated a few devices based on the exfoliated BiSbTeSe₂(BSTS) flakes and performed systematic transport measurements. The authors used three approaches to acquire the size of the hybridization gap in these devices. By combining the DFT calculations, the authors claimed the observation of the quantum spin Hall effect in a 10QL device. The story is interesting and the paper is written well.

Response: We thank the reviewer for appreciating this work.

However, I have two major concerns about the claim of this paper.

(1) As described on Page 3 and shown in Fig. 1a, the theoretical calculations clearly show that the BSTS hybridization gap exhibits an oscillatory pattern between the QSH and normal insulators. In other words, the quantum spin Hall phase appears in the devices within a broad thickness range. However, why the quantum spin Hall phase appears only in the 10QL but disappears in the devices with other thicknesses (e.g. 8 and 9 QL)? I understand that the electric field can tune the devices with different thicknesses into the QSH phase regime easily, as shown in Fig. 4.

Response: We agree with the reviewer that we are still missing the evidence for the oscillatory pattern between the quantum spin Hall and normal insulators. We added a discussion for the possible reason for not seeing the QSH phase in the second inverted regime (5-6QL) in the revised manuscript. A possible reason is the dramatic decrease in sample quality as the thickness of the hBSTS reduces. For example, our 7 and 6 nm hBSTS show significantly lower mobility than the thicker hBSTS and no quantum Hall development at high magnetic field. This significantly reduced electron mean free path can obscure the transport signature of the QSH phase. In such cases one will have to reduce channel lengths to search for the QSH state in the second inverted regime. Also, as the thinner hBSTS is a large gap semiconductor, the contact issue can make it harder to realize the QSH phase.

(2) How many 10QL devices show the quantum spin Hall effect? If there is only a 10QL device showing the QSH phase. The longitudinal resistance close to $h/2e^2$ might be observed by chance.

Response: We understand the reviewer's concern about the longitudinal resistance close to $h/2e^2$ being observed by chance. We emphasize that the evidence of the QSH phase in our 10QL hBSTS device is not limited to the observation of the longitudinal resistance close to $h/2e^2$, but further supported by the two-terminal and non-local transport data as shown in supplementary information Fig. S7 in the revised manuscript.

In addition, we have recently reproduced the results in a second 10QL hBSTS device which clearly shows the longitudinal resistance close to $h/2e^2$ with additional two-terminal resistance data shown in the supplementary information Fig. S8 (also shown here) in the revised manuscript.

Below are my minor concerns:

(3) The 2D-3D TI crossover thicknesses are 6 QL, 4 QL, and 2 QL for Bi₂Se₃, Sb₂Te₃, and Bi₂Te₃, respectively (Ref. 7; PRL 108, 016401(2012); Adv. Mater. 22, 4002(2010)). This crossover thickness should be independent of the electrical properties of the TI films, so I don't quite understand why the 2D to 3D TI crossover thickness of BSTS in this work is much larger. It would be great that the authors should list the parameters used in their DFT calculations and add more discussions about this issue in the paper.

Response: We added more discussions about the larger 2D-3D TI crossover thickness observed in this work in the revised manuscript. We agree with the reviewer that the thickness for the 2D-3D TI crossover should be independent of the electrical properties of the TI films. However, we argue that this crossover due to the hybridization between the top and bottom surface states is not an abrupt process. Our point is made very nicely by the theoretical band structure calculations in the case of Bi₂Se₃ [New J. Phys. 16, 063022 (2014)], where the hybridization gap can be traced beyond 10QL. However, these gap sizes are smaller than the resolution limit of most experimental techniques e.g. angle-resolved photoemission spectroscopy. Some evidence of these sub-meV hybridization gaps is provided in phase coherent transport measurements of 12-17 nm Bi₂Se₃ [Nat. Comm. 4, 2040 (2013)]. In conclusion, there is not a hard cutoff for the crossover thicknesses but is contingent upon resolution.

(4) In Fig.3b, the vertical axis is in the log scale, please change it to the linear scale because the log scale will hide the R_{xx} quantization of the QSH phase. Based on the data shown in Fig.4a, I speculate that R_{xx} is larger than $h/2e^2$ but there is no plateau feature by tuning the gate.

Response: We added the Fig. 3b in a separate linear-scale plot in supplementary information Fig. S7 together with the non-local transport data. As shown in the figure, by controlling the gate voltage into the hybridization gap, the R_{xx} peaks very close to a value of $h/2e^2$. We also provide the two-terminal and non-local resistances measured on the same device, which show the expected resistance values according to the Landauer-Buttiker formalism for helical edge conduction channels. The plateau feature in gate voltage is better resolved in the two-terminal resistance versus gate voltage plot.

(5) The estimated inverted gap for the 10 QL BSTS device is ~ 20 meV. This gap size corresponds to the critical temperature of the QSH phase ~ 200 K, which is much higher than the ~ 40 K reported in this paper. The authors need to add more discussions about this issue in the paper.

Response: We thank the reviewer for pointing out the mismatch between the inverted gap size and the critical temperature for the QSH phase. We clarify that the inverted gap estimated for the 10QL hBSTS is ~ 8 meV (the gap size for 9 QL hBSTS is ~ 20 meV). One of the big issues with the QSH effect is scattering between the helical edges. Given that the channel size is larger than the mean free path, the scattering effect can dominate at high temperature, thus limiting that critical temperature. We added a discussion to the revised manuscript.

(6) The caption of Fig.2 should include the device information.

Response: We added the device information in Fig. 2 caption in the revised manuscript.

Based on my above comments, I will not recommend this paper for publication in Nature Communications. If the authors could reproduce this phenomenon in more devices, I will reconsider this work in the future.

Response: We hope the additional data provided in this revision (second 10QL device, non-local data in multiple devices, linear plots, etc) will convince the reviewer of the two-dimensional topological phase in hybridized three-dimensional topological insulators. We appreciate the reviewer's consideration.

Reviewer #4 (Remarks to the Author):

Dear Editor,

after reviewing the manuscript “Emergent Helical Edge States in a Hybridized Three-Dimensional Topological Insulator”, I remain unconvinced that the paper supports the conclusions of any new physics. In addition, I find the manuscript to be quite poorly written, and very hard to follow.

This impression starts in the introduction: For example, the choice of Reference 2 and 3, which are neither the firsts, nor the most well-known papers for observations of magnetic TI's suggest a complete unfamiliarity with the field, as does the fact that the authors appear to be unaware that both 2D and 3D magnetic TIs have been studied, and indeed the transition between the two is discussed in the literature. This of course can be repaired with a proper rewriting of the introduction. The significant issue however is that the scientific evidence as presented is simply not compelling.

Response: We decided to remove the reference to magnetic 3D TIs as it is not related to the subject of the manuscript. The only reason we had included it previously was to make a comparison of the less well-studied hybridization gap with the exchange gap. Since magnetic TIs are extraneous to the current study, we decided to remove the reference from our manuscript.

Regardless we note that this manuscript is not about magnetic topological insulators and therefore we did not see the need to cite the history of magnetic TIs. We are well familiar with the field of magnetic topological insulators and we find this characterization by the referee extremely unfair. We also note that all other referees explicitly mention that this is a well-written manuscript.

The main claim of the paper relies on observing the helical edge channels under certain conditions (either intrinsically in a 10nm thick sample, or by applying large electric fields to a 9nm thick sample). This claim is based on R_{xx} having a value close to $h/2e^2$, but that could be merely coincidental. To convincingly show this results from edge states, one should use various non-local measurement configurations and show with proper quantized plateau values consistent with the Landauer-Buttiker formalism for two helical edge channels in each configuration. This can easily be done on their existing samples, as they have a Hall bar geometry, which offers multiple possible configurations with distinct expected plateau values.

Response: We thank the reviewer for suggesting non-local measurement to further confirm the helical edge states observed in our 10nm thick hBSTS. We have added two-terminal and non-local transport data in the supplementary information Fig. S7 (also shown here) in the revised manuscript. We show that the two-terminal and non-local resistances measured in different configurations agree with the expected values for helical edge states derived from the Landauer-Buttiker formalism.

In addition to this main criticism, here is a list (perhaps incomplete, as the paper is very hard to follow) of additional significant issues which need addressing:

- There is a discrepancy between the gap size obtained from the Arrhenius fit, and the other two methods (Fig. 2d), by about a factor of 2-3. Why?

Response: We clarify that the thermal activation gap often suffers due to the smearing effect which results in a typically small measured value in comparison with the actual gap size. We note that other works in the literature also make the same observation [see e.g. Phys. Rev. B 99, 201101 (2019)].

- What justifies the use of the Arrhenius fit, over for example the variable range hopping. It is clear in the inset to Fig. 2a that the G_{xx} dependence is not linear for a full range of $1/T$. Indeed, with the data for Fig. 2a covering less than a decade, any attempt to extract exponential/power law behaviour is fraught with danger.

Response: We agree with the reviewer that the non-linear G_{xx} at low temperature can be a result of variable range hopping. To justify the Arrhenius fit region, we show here the G_{xx} versus $1/T$ plot for the 9 nm hBSTS, where the bulk activation, surface activation and variable range hopping regions are identified. We recalculate the activation gap in the appropriate region with the Arrhenius formula suggested by one of the reviewers in the revised manuscript. We also agree with the reviewer that the activation gap estimated from the data covering less than a decade, especially for thicker samples, can be inaccurate. Therefore, we also used two other methods (differential conductance and quantum capacitance) to extract the hybridization gap size.

- The assignment of the arrows in Fig. 2b (and for the other samples in Fig.S5) seems rather arbitrary, and this affects the extracted gap using this method.

Response: We clarify that the arrows in Fig. 2b are not an arbitrary assignment but based on the literature [see for example the paper Phys. Rev. Lett. 98, 206805 (2007) cited over 4000 times]. We thank the referee for recognizing that this might seem arbitrary and therefore need elaboration. To better illustrate it, we show the extraction of the gap size from the dI/dV curves based on the two-slopes method as described in the supplementary information and Fig. S5 in the revised manuscript.

- How can they be sure of the quality of their contacts. The argument (found in the supplement) that they check them by additionally measuring at different gate voltages where the sample is conducting is not convincing, as the contact quality can also vary significantly with gate voltage (especially when the sample approaches its insulating gap.)

Response: We agree with the reviewer that it is conceivable that the contact quality can be an issue in the gap in particular in the thinner devices with gap sizes being hundreds of meV. Thus, we use multiple techniques to measure the gap. We note though that in the case of 10 nm sample with ~ 8 meV gap this should not be an issue. Additionally, we note that we perform dI/dV measurements of all our devices which provides a good indicator of the bad contact issue. For example, the data below where the dI/dV dip at zero bias is preserved even when outside the gap, is one such bad contact device. We exclude bad contact devices from the analysis as they do not give the actual gap size.

- It is quite hard to tell for certain how Fig. 3d (the color map) would look like without the dashed lines, but I suspect they are misleading and trying to guide the readers eye to see something that is not really visible in the data itself.

Response: To clarify the doubt of the reviewer, we have taken the dashed lines off (shown in the figure below) for the reviewer's benefit. Also, the line profile of G_{xx} as provided in the original manuscript resolves the same feature better without any guide to the eye.

-The G_{xx} magnetic field sweep in Fig. 3c does not look numerically processed or like a poorly resolved measurement. Moreover, they say in "methods" that they current bias the sample at 10-20 nA and AC 17.777 Hz. Doing current biased measurements on an insulator is not proper and can lead to measurement artefacts. A better measurement of this G_{xx} signal might reveal similar features to the ones observed in Fig. 3d marked with red arrows (that are interpreted as band inversion, or destruction of helical edge states). This should be checked.

Response: It is not clear to us why the reviewer says the data in Fig. 3c is numerically processed or poorly resolved. The conductance is clearly a fraction of e^2/h . We are aware that there are better methods than current bias to probe an insulator. However, the $\sim 0.1 e^2/h$ (~ 250 kOhm) is still much smaller than the input impedance (10 MOhm) of the lock-in amplifier. As a result, current bias is a pretty good

measurement in this regime. Also, we would like to note that in our original ‘methods’ section, in addition to current bias, we also listed the methods of voltage bias differential conductance and quantum capacitance as other measurements that we used in our work. We would also like to point out that the reviewer’s comment, that a better measurement of the 9 nm sample might reveal similar features as the 10 nm sample, is inapplicable because the resistance value of the 9 nm sample in Fig. 3c does not go anywhere close to h/e^2 which is required for gap closing and band inversion (gap reopening).

- The 10nm sample that has $\sim h/2e^2$ R_{xx} resistance appears to be a single device. All thinner samples are insulating. Is the result reproducible?

Response: The result for the 10 nm hBSTS is reproducible. We have shown the temperature and gate dependent data for an additional 10 nm hBSTS sample in the supplementary information (also shown here) in the revised manuscript.

- In addition, there is one sample in the supplement labelled as "10nm (no gap)". It is unclear that this means. Is there so much sample to sample variations? How robust are then any of the observations?

Response: We clarify that the “10 nm (no gap)” sample is a 10 nm sample where neither thermal activation nor dI/dV gap was observed. We listed this sample because we are being exhaustive so as to include all the devices we measured. Indeed, there is likely to be sample to sample variation when performing exfoliation from bulk crystal since the crystal can have some inhomogeneity. This device appears to be metallic, possibly due to bulk conduction which is a common problem in 3D topological insulators.

-How can it be excluded that the large electric field are affecting the bulk, and activating bulk carriers?

Response: We agree with the reviewer that the large electric field can also shift the bulk band. However, as the bulk band gap is significantly larger than the surface gap [Nat. Phys. 6, 584-588 (2010)], the electric field tuning of the bulk gap is rather small. The less effective electric field tuning in thinner

hBSTS with larger hybridization gap also supports the conjecture that the applicable electric field range via the solid state gate is incapable of controlling bands far from the Fermi level.

- In Fig. 4g where by applying large electric fields the authors claim the transition to the helical edge channel regime, the ρ_{edge} does not look like it will saturate with electric field strength, but looks like it would drop even further with larger field. This would contradict the entire interpretation. In this picture, one could simply adjust the maximum field to produce a different value than the observed $\sim h/2e^2$ is one wished.

Response: We clarify that the swept range of the gate voltage, and thus the maximum field that can be applied, depends on the breakdown strength of the gate dielectric. Pushing the gate voltage too much can cause dielectric breakdown and gate leakage. We take precautions when sweeping large gate voltage to prevent irreversible dielectric damage, and we do not intentionally stop sweeping the gate upon achieving a particular value. In fact, we show here the last cooling of the 9 nm sample with a higher positive displacement field applied, which does not present a further drop or monotonic decrease in the resistance at large displacement field. Note that the resistance at negative displacement fields changed due to thermal cycling, however, the positive displacement fields preserve the same feature as the previous cooling.

- There appears to be some confusion between ρ_{xx} and R_{xx} in Fig. 4a-d (between the labels on the axis and the caption).

Response: We have corrected the typo in the caption of Fig. 4.

-The manuscript also contains unclear statements like "cleaner surface bands". This reader has no idea what that should mean?

Response: We apologize for using a statement like "cleaner surface bands" without explanation. We have removed it in the revised manuscript.

In summary, the current data does not provide any compelling evidence of the authors claim, and until that is resolved, I cannot recommend this paper for publication.

Response: We hope that, after our careful consideration and changes in response to the reviewer comments, our revised version addresses the critiques of the reviewer satisfactorily. The authors thank the reviewer for his/her comments.

REVIEWER COMMENTS

Reviewer #2 (Remarks to the Author):

I have read the revised manuscript, and the response to the referee report. The revised manuscript now includes a new representative device of 12 nm and a “quantize-like” device of 10 nm. While I appreciate the authors’ efforts in improving this work, I cannot recommend publishing the manuscript in the existing form because of the following unsolved problems:

1) The new 10-nm device S3 (Fig. S8) shows results away from quantization, which did not demonstrate the quantization, but further demonstrated the “non-quantization”. Technically, I can understand that it is challenging to fabricate a “quantized” device as the S1, since many of the fabrication process could induced the disorders, make the results not quantized. I would suggest the authors tone down the “quantization” claim, and present the measurement results with more measurement configuration geometry (not just 2 or 3 pairs, but try to list most of the configurations) based on the hall bar. If the data shows consistency with the Landauer-Buttiker formalism, this would be a convincing result.

2) The mix presentation of R_{xx} and ρ_{xx} , e.g., Fig. S3, Fig. S11, and Fig. S12 shows ρ_{xx} , while others show R_{xx} . This confused me a lot in understanding the data. As this confused me a little bit, the following point is based on $R_{xx} = \rho_{xx}$. In Fig. S11b, which is 10 nm hBSTS (S1?), supposed to be quantized (~12.9 kOhm), shows a resistance > 15 kOhm, much larger than the quantization, inconsistent with Fig. 3 and Fig. S7a.

Others to further improve the manuscript:

1) The monolayer WTe₂ could be categorized to the monolayer 1T'-TMD (actually it is a Td-TMD on real crystal, the monolayer form of monolayer Td-WTe₂ and 1T'-WTe₂ are the same.)

2) Table S2 should include the information of the new 12 nm device.

3) What is the layer thickness of the hBSTS? The authors should correlate the layer thickness to the quintuple layers (QL) for the general audience.

Reviewer #3 (Remarks to the Author):

I read the response letter to the comments from the four reviewers and the revised manuscript carefully. The authors have addressed all my concerns and I am satisfied with their responses. I think this paper can be published in Nature Communications.

Reviewer #4 (Remarks to the Author):

I find it interesting that the authors complain that my comment of the manuscript being poorly written is unfair, but then essentially agree with me on each point I raise to that effect. Perhaps we just have a very different view as to what constitutes a well written manuscript.

In any event, the paper is significantly improved. In particular, the inclusion of non-local measurements in the supplemental significantly strengthens their claims. I would prefer to have these shown more prominently in the main text, but as long as they are available to the reader, in the end, this is satisfactory. This, together with the inclusion of data on additional devices now, in my opinion, has the manuscript meeting the threshold of justifying the scientific claims.

I have to admit that I am personally still unconvinced that the paper meets the general/broad interest requirements of a Nature communications, but in its revised form, I find that it should be published somewhere. If the Editors feel this journal is the appropriate place, I certainly would not object, but if it was my decision, I would probably place it in a somewhat more specialized journal.

Responses to Reviewers' comments:

Reviewer #2 (Remarks to the Author):

I have read the revised manuscript, and the response to the referee report. The revised manuscript now includes a new representative device of 12 nm and a “quantize-like” device of 10 nm. While I appreciate the authors' efforts in improving this work, I cannot recommend publishing the manuscript in the existing form because of the following unsolved problems:

1) The new 10-nm device S3 (Fig. S8) shows results away from quantization, which did not demonstrate the quantization, but further demonstrated the “non-quantization”. Technically, I can understand that it is challenging to fabricate a “quantized” device as the S1, since many of the fabrication process could induced the disorders, make the results not quantized. I would suggest the authors tone down the “quantization” claim, and present the measurement results with more measurement configuration geometry (not just 2 or 3 pairs, but try to list most of the configurations) based on the hall bar. If the data shows consistency with the Landauer-Buttiker formalism, this would be a convincing result.

Response: We appreciate the reviewer's understanding of the challenges involved in fabrication of the TI devices. Indeed, the fabrication process can induce disorders in the sample and prevent the observation of quantization. We also emphasize that the exfoliated TI samples are not as robust as epitaxially grown TIs, for example. Prolonged exposure of the exfoliated surface to the ambient can also degrade the sample. Our fabricated TI devices with hBN/graphite encapsulation provide better protection to the sample. We note that the new 10-nm device S3 showed results closer to the quantized value in its first cooldown (see the figure below). Subsequently, we tried (unsuccessfully) to transfer a hBN/graphite capping layer on this sample. As the device was exposed to ambient, the more complete set of data acquired in the subsequent cooldown (presented in the previous revision, SI Fig. S8) showed further reduced in-gap resistance. We attribute this change to a shift in the chemical potential of the top surface, causing further deviation from quantization. However, the weak and saturating temperature-dependent resistance was still observed. As the device condition has changed, further measurements in additional cooling cycles no longer give further supportive results. Given this scenario, we agree with the reviewer to tone down the quantization claim and rephrase it with 2D topological states. We have also listed all the configurations measured for the 10 nm devices S1 and S3 in Table S4 in the revised manuscript and below for the reviewer's reference.

Devices	Configurations	R_{exp} (k Ω)	R_{theory} (k Ω)
10 (S1)	$R_{14,23}$	12.8 ± 0.4	12.906
	$R_{12,12}$	21.9 ± 0.4	21.150
	$R_{13,13}$	35 ± 0.5	34.416
	$R_{35,12}$	8.8 ± 0.3	8.604
10 (S3)	$R_{14,23}$	12.2 ± 0.1	12.906
	$R_{12,12}$	20.8 ± 0.3	21.150

2) The mix presentation of R_{xx} and ρ_{xx} , e.g., Fig. S3, Fig. S11, and Fig. S12 shows ρ_{xx} , while others show R_{xx} . This confused me a lot in understanding the data. As this confused me a little bit, the following point is based on $R_{xx} = \rho_{xx}$. In Fig. S11b, which is 10 nm hBSTS (S1?), supposed to be quantized (~ 12.9 k Ω), shows a resistance > 15 k Ω , much larger than the quantization, inconsistent with Fig. 3 and Fig. S7a.

Response: We apologize for the confusion created by our use of both R_{xx} and ρ_{xx} in the S.I. figures. We clarify that the ρ_{xx} is calculated from the measured R_{xx} using the geometry factor. To clarify the data, we changed the relevant plots to R_{xx} in the revised manuscript. We note that for Fig. S11b, R_{xx} is much closer to the quantized value. The slightly higher apparent value near 0T (bold red line) as pointed by the blue arrow is due to the remnant field as the data were taken while sweeping both gate voltage and magnetic field simultaneously (“0T” near the peak is in essence halfway between 0T and 0.45T).

Others to further improve the manuscript:

1) The monolayer WTe₂ could be categorized to the monolayer 1T'-TMD (actually it is a Td-TMD on real crystal, the monolayer form of monolayer Td-WTe₂ and 1T'-WTe₂ are the same.)

Response: We thank the reviewer for clarifying the structure of the monolayer WTe₂. We grouped the WTe₂ to the monolayer 1T'-TMDs in the revised manuscript.

2) Table S2 should include the information of the new 12 nm device.

Response: We included the information about the new 12 nm device in the revised manuscript.

3) What is the layer thickness of the hBSTS? The authors should correlate the layer thickness to the quintuple layers (QL) for the general audience.

Response: We added the layer thickness in the main text, page 3, in the revised manuscript.

Reviewer #3 (Remarks to the Author):

I read the response letter to the comments from the four reviewers and the revised manuscript carefully. The authors have addressed all my concerns and I am satisfied with their responses. I think this paper can be published in Nature Communications.

Response: We thank the reviewer for their constructive comments and for being positive about this work.

Reviewer #4 (Remarks to the Author):

I find it interesting that the authors complain that my comment of the manuscript being poorly written is unfair, but then essentially agree with me on each point I raise to that effect. Perhaps we just have a very different view as to what constitutes a well written manuscript.

In any event, the paper is significantly improved. In particular, the inclusion of non-local measurements in the supplemental significantly strengthens their claims. I would prefer to have these shown more prominently in the main text, but as long as they are available to the reader, in the end, this is satisfactory. This, together with the inclusion of data on additional devices now, in my opinion, has the manuscript meeting the threshold of justifying the scientific claims.

Response: We thank the reviewer for their constructive comments and suggestions that helped improve the quality of this manuscript.

I have to admit that I am personally still unconvinced that the paper meets the general/broad interest requirements of a Nature communications, but in its revised form, I find that it should be published somewhere. If the Editors feel this journal is the appropriate place, I certainly would not object, but if it was my decision, I would probably place it in a somewhat more specialized journal.

Response: We thank the reviewer for being positive about the revised version of the manuscript. We would like to clarify the importance of this work by listing the following points: (i) This work provides the first evidence that a 2D topological insulator can exist in the hybridization regime of a 3D topological insulator. In addition to joining a handful of 2D TI materials, our work provides a blueprint for realizing this highly sought-after state in other 3D topological insulators. (ii) This work provides understanding of electric- and magnetic-field-induced topological phase transitions, which can be generalized to different topological systems, (iii) This work demonstrates excellent tuning capability between trivial and topological phases in hBSTS-based van der Waals heterostructures, which can be useful for topological quantum electronics. Together with the extensive energy gap data we provide using several complementary techniques as a function of thickness, temperature and electric/magnetic fields, we believe our work will be of interest to a broad community of researchers.

REVIEWERS' COMMENTS

Reviewer #2 (Remarks to the Author):

I have read the revised manuscript and the response to the referee report. The authors explained the non-quantization data of device S3 due to degradation after fabrication, which makes sense. The authors also present nearly quantized data for the first cooldown, demonstrating the quantization features. I appreciate the honesty in presenting the data. Since the authors keep claiming the quantization of $h/2e^2$ throughout the paper, an independent set of "quantization" data is necessary to support such a claim. I would suggest the authors add the first cooldown data into Fig. S8, and explaining the reasons for the decreasing of the resistance in the second run. For the monolayer WTe₂, I would suggest noting 2D TI due to Nature Physics 13, 677 (2017) and Science 359, 76 (2018).

After the minor revision, the manuscript should be good for publication.

Responses to Reviewers' comments:

Reviewer #2 (Remarks to the Author):

I have read the revised manuscript and the response to the referee report. The authors explained the non-quantization data of device S3 due to degradation after fabrication, which makes sense. The authors also present nearly quantized data for the first cooldown, demonstrating the quantization features. I appreciate the honesty in presenting the data. Since the authors keep claiming the quantization of $h/2e^2$ throughout the paper, an independent set of "quantization" data is necessary to support such a claim. I would suggest the authors add the first cooldown data into Fig. S8, and explaining the reasons for the decreasing of the resistance in the second run.

Response: We thank the reviewer for their suggestion and have added the first cooldown data into Fig. S8, together with the explanation of the reasons for decrease of the resistance in the second run.

For the monolayer WTe₂, I would suggest noting 2D TI due to Nature Physics 13, 677 (2017) and Science 359, 76 (2018).

Response: We noted the 2D TI for monolayer WTe₂ in the Table S4 in the revised manuscript.

After the minor revision, the manuscript should be good for publication.

Response: We appreciate the reviewer for all the constructive comments which helped a lot in improving the quality of the manuscript.